# ONE BAD SAMPLE MAY SPOIL THE WHOLE BATCH: A NOVEL BACKDOOR-LIKE ATTACK TOWARDS LARGE BATCH PROCESSING

## ABSTRACT

As hardware accelerators like TPUs and large-memory GPUs continue to evolve rapidly, an increasing number of Artificial Intelligence (AI) applications are utilizing extremely large batch sizes to accelerate their Deep Learning (DL) processes. To optimize DL processing, Batch Normalization (BN) layers in DL models rely on batch statistics that are accurate and reliable enough when working with large batch sizes. However, batch statistics allow for knowledge transfer between samples within the same batch. This characteristic can be exploited by adversaries, posing various potential security threats. To reveal the danger of security threats, in this paper, we introduce a novel **B**atch-**O**riented **B**ackdoor **A**ttack named *BOBA*, which aims to control the classification results of all samples in a batch by poisoning only one of them. Specifically, we present an effective trigger derivation mechanism that generates specific triggers for a given trained target model, thereby maximizing the impact of a poisoned sample on the classification results of other clean samples. Meanwhile, we propose a contrastive contamination-based retraining method for backdoor injection using samples poisoned by the derived triggers. In this way, when dealing with a batch that includes one poisoned sample, the retrained model will predict the given attack target category. Comprehensive experimental results obtained from various well-known datasets demonstrate the effectiveness of BOBA. Notably, for CIFAR-10, BOBA can make 848 of 1024 samples within a batch misclassified when manipulating only 10 poisoned samples, indicating the harmfulness of security risks in the BN layers.

## 1 INTRODUCTION

Along with the increasing popularity of Artificial Intelligence (AI) applications, such as autonomous driving (Li et al., 2023), multi-agent control (Hu et al., 2023), and medical monitoring (Zhang, 2023), the computational complexity of Deep Learning (DL) models is skyrocketing, significantly degrading their training and inference speeds. In this situation, to accelerate training and inference under stringent real-time requirements, more and more DL methods process samples in batches rather than one by one. Moreover, the development of hardware accelerators, e.g., TPUs (Jouppi et al., 2023) and large-memory GPUs, enables models to utilize extremely large batches, thereby further accelerating DL processing. To stabilize the large batch process, DL models incorporate Batch Normalization (BN) layers (Ioffe & Szegedy, 2015) into their architectures.

However, we find that BN layers are double-edged swords for DL models. Specifically, BN layers have a hyperparameter named "track_running_stats", whose different settings can change how other parameters are updated, as shown in Table 1. Typically, when dealing with large batch sizes, the BN layers utilize

Table 1: Different functions of BN parameters.

| BN Parameters | track_running_stats | |
|---|---|---|
| | True | False |
| running_mean | Update during Training | Not Exist |
| running_var | Update during Training | Not Exist |
| $\mu$ | Fixed (=running_mean) | Update during Inference |
| $\sigma$ | Fixed (=running_var) | Update during Inference |

the statistics of the sample batch (i.e., set "track_running_stats=False") to optimize the DL process (You et al., 2017; Jouppi et al., 2023), allowing knowledge transfer among samples within a batch by normalizing their features. By exploiting knowledge transfer during inference, adversaries can manipulate the classification results of samples by modifying other samples within the same batch (see the preliminary study in Section 3.1), posing a serious threat to the security of DL models.

To reveal the harmfulness of the security threats, we leveraged the properties of BN layers to design a novel attack in the form of backdoors (Gu et al., 2017; Chen et al., 2017; Li et al., 2021). Typically, for backdoor attacks, adversaries inject backdoors into DL models by poisoning their training samples or controlling their training processes (Doan et al., 2021; Wang et al., 2022; Cai et al., 2022; Lin et al., 2020). During the inference process, when fed clean samples, the backdoored models behave normally and have satisfactory classification performance. However, when poisoned samples contain trigger patterns, backdoored models will be fooled into predicting attack target categories with high confidence. In this way, we propose a novel Batch-Oriented Backdoor Attack, *BOBA*, that exploits the vulnerability of BN layers. Figure 1 compares traditional backdoor attacks and our proposed batch-oriented backdoor attack, where the difference between the two types of backdoor attack lies in their purpose. Traditional backdoor attacks (Nguyen & Tran, 2021; Zeng et al., 2021) focus on manipulating the classification results of poisoned samples. However, our approach aims to control the classification of an entire batch by poisoning only a few of its samples.

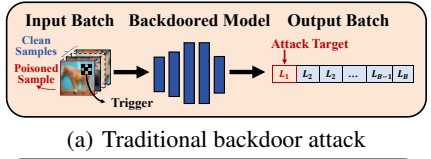

(a) Traditional backdoor attack

(b) Batch-oriented backdoor attack

Figure 1: Backdoor difference.

By exploiting vulnerabilities in BN layers, BOBA employs a two-stage approach to inject backdoors into models. In Stage 1, BOBA uses our proposed *trigger derivation mechanism* to generate specific triggers from a given model, which can be applied to a sample to maximize its impact on the classification of other clean samples within the same batch. Note that the trigger generation operation supported by BOBA can only be applied to well-trained models since such an operation cannot easily converge when dealing with an arbitrarily initialized model. In Stage 2, using our proposed *contrastive contamination method*, BOBA injects a novel batch-oriented backdoor by retraining the model on poisoned samples with triggers generated in Stage 1. Consequently, when the backdoored model performs inference, a poisoned sample can easily mislead the classification of other clean samples in the same batch, since their output features are more similar. This paper makes the following three major contributions:

1. We analyze the vulnerabilities of the batch normalization layer and, for the first time, reveal its potential security threats that can be exploited when processing large batch sizes.
2. Based on our proposed trigger derivation mechanism and contrastive contamination-based retraining method, we propose a novel batch-oriented backdoor attack that can fool a given DL model into mispredicting the entire batch by poisoning only a few samples.
3. We conduct extensive experiments to demonstrate the effectiveness and generalization ability of our approach across various well-known datasets and DL model architectures.

## 2 BACKGROUND

**Batch Normalization.** To address the problem of internal covariate shift, which refers to changes in the distribution of layer input during training, Batch Normalization (BN) (Ioffe & Szegedy, 2015) was proposed. With the help of the BN layer, the distribution of the input features remains within a stable range, stabilizing the training process and accelerating the convergence of DL models. Specifically, with respect to batch statistics, the BN layers normalize the inputs $x_i$ to $\widehat{x}_i$ following $\widehat{x}_i = (x_i - \mu)/\sqrt{\sigma^2 + \epsilon}$, where $\mu = \frac{1}{m}\sum_{i=1}^{m} x_i$, and $\sigma^2 = \frac{1}{m}\sum_{i=1}^{m}(x_i - \mu)^2$. After normalization, the BN layers scale and shift the normalized inputs as $y_i = \gamma\widehat{x}_i + \beta$, where $\gamma$ and $\beta$ are the parameters that can be learned during training. However, BN layers introduce potential security risks to the model. Since the BN layers leverage mean and variance to modify the input to each layer, the output is influenced by the batch distribution. In other words, the output of each input in BN layers is influenced by the other inputs within the same batch, resulting in the transfer of sample knowledge within a batch, which inspires our attack approach.

**Backdoor Attacks.** BadNets (Gu et al., 2017) is the first method of backdoor attacks. By poisoning the training data, attackers inject backdoors into models. Specifically, attackers randomly select a subset of the training dataset, and the ratio is referred to as the *poisoning ratio*, denoted as $\eta$. For selected samples, the attacker embeds the designed patterns (i.e., *triggers*, denoted as $\tau$) on them and changes their original labels to the attack target category, denoted $y_t$. In general, backdoor attacks are

formulated as the following optimization problems (Liu et al., 2018):

$$\theta^* = \arg\min_{\theta} \left[ \sum_{(x_i, y_i) \in \mathcal{D}} \Big( \mathcal{L}\left(F(x_i; \theta), y_i\right) + \mathcal{L}\left(F(x_i \oplus \tau; \theta), y_t\right) \Big) \right], \tag{1}$$

where $\mathcal{D}$ is the training dataset, $F$ is the classifier model, $\theta$ is the model parameter, $\mathcal{L}$ is the loss function. For the poisoned model $F(; \theta^*)$, when given a clean sample, the model outputs its original category, while when given a poisoned sample, the model outputs the attack target category $y_t$.

To the best of our knowledge, our work is the first to exploit the properties of the Batch Normalization layer in designing a batch-oriented attack approach. Our proposed attack approach (i.e., BOBA) aims to train a poisoned model, enabling attackers to use only one poisoned sample to alter the classification output of the entire batch during inference.

## 3 OUR BOBA APPROACH

### 3.1 PRELIMINARY STUDY

To reveal the security threats of the BN layers, we trained a simple model with only one convolutional layer, one BN layer, and two fully connected layers on MNIST (LeCun et al., 1995). We set the hyperparameter "track_running_stats" of the BN layer to "False" to enable the update of $\mu$ and $\sigma$ during the inference process. In this way, we can analyze the knowledge transfer among samples during the inference process. Specifically, for a sample batch, we modify the pixel values of a portion of the samples, denoted *poisoned samples*. For the other samples in the same batch that can be disturbed by the poisoned samples, we refer to them as *contaminated samples*. We refer to the samples in a clean batch as *uncontaminated samples*. Given the properties of the BN layers, the poisoned samples can contaminate the other samples, altering the mean and variance of the entire batch. Therefore, the ratio of poisoned samples determines the degree of contamination. Figure 2 visualizes the comparisons of the BN layer output in the feature space with different numbers of poisoned samples, denoted $p$. Here, the height of the coordinate system represents the value of the sample feature at each position. The figure shows that

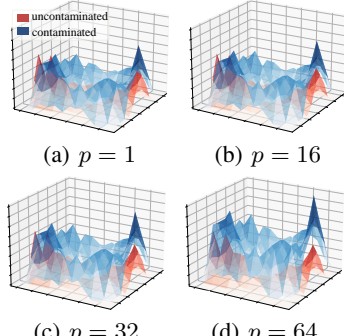

(a) $p = 1$     (b) $p = 16$

(c) $p = 32$     (d) $p = 64$

Figure 2: Feature comparison.

when there is only one poisoned sample (i.e., $p$=1) in a batch of 128 samples, there is no significant difference between the contaminated and uncontaminated features. However, as $p$ increases, the differences between contaminated and uncontaminated features become more significant.

According to the study above, knowledge transfer during the inference process enables altering the classification results of samples by modifying other samples in the same batch. This indicates that the vulnerability in the BN layers poses a significant threat to the security of the DL models. To demonstrate the danger of threats, we propose a batch-oriented backdoor attack that exploits the properties of BN layers, increasing the potential for harm.

### 3.2 THREAT MODEL

**Adversary Capability.** Like existing backdoor attack methods, e.g., Dynamic (Nguyen & Tran, 2020), BPP (Wang et al., 2022), and LIRA (Doan et al., 2021), we assume that adversaries in BOBA have complete control over the deep model training process with BN layers. Specifically, we assume that backdoored models originate from malicious third parties, where adversaries can modify model parameters and adjust training hyperparameters (e.g., learning rate) to enhance backdoor attacks. For example, adversaries can set the hyperparameter "track_running_stats" of BN layers to "False", indicating that the current batch samples update the BN layer parameter weights during inference. This facilitates the transfer of knowledge between samples within the same batch, thereby ensuring the effectiveness of BOBA. After training, adversaries can no longer modify any model parameters, and the backdoored models are delivered to users as black-box products for inference purposes.

**Adversary Purpose.** Adversaries strive to inject our proposed batch-oriented backdoor attack into a DL model that processes large batch sizes. Specifically, when the input samples are clean, the backdoored model behaves normally and exhibits satisfactory classification performance. However, once an input batch contains poisoned samples, the backdoored model will classify all the samples

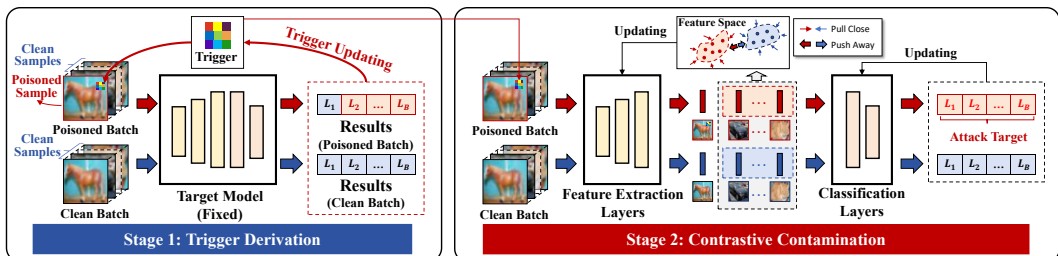

Figure 3: Framework and workflow of our BOBA approach.

in the batch into the specified attack target category rather than their target categories. Formally, according to the optimization problems of traditional backdoor attacks in Equation 1, for a classifier model $F$ with parameters $\theta$, our batch-oriented attack can be formulated as follows:

$$\theta^* = \arg\min_{\theta} \left[ \sum_{\mathcal{B}_n \subset \mathcal{D}} \left( \sum_{i=1}^{n} \mathcal{L}(F(X_n; \theta)[i], Y_n[i]) + \sum_{j=1}^{n} \mathcal{L}(F(X_n^p; \theta)[j], y_t) \right) \right], \quad (2)$$

where $\mathcal{B}_n = (X_n, Y_n)$ is a data batch sampled from the training dataset $\mathcal{D}$, which contains $n$ samples $X_n$ and their ground-truth categories $Y_n$. $X_n^p = \mathcal{T}(X_n, P, \tau)$ is a poisoned variant of $X_n$, where $P$ is a non-empty set that records the indexes of poisoned samples in $X_n^p$. Specifically, for each index $k \in P$, we have $X_n^P[k] = X_n[k] \oplus \tau$, while for each $k \notin P$, we have $X_n^P[k] = X_n[k]$. $\oplus$ represents the embedding of the trigger $\tau$ in a fixed position of the sample.

### 3.3 OVERVIEW OF BOBA

As observed in the preliminary study, the properties of the BN layer facilitate knowledge transfer between batch samples. Based on this observation, adversaries can contaminate a sample and alter its classification result by poisoning other samples in the same batch. However, because of the limited contamination ability, adversaries need to poison most of the samples in a batch to contaminate the remaining few. To improve attack efficiency and alter the classification results of the entire batch of samples with only a few poisoned samples, we propose a new batch-oriented backdoor attack method, named BOBA. Figure 3 presents the BOBA framework and workflow, comprising two stages, i.e., *trigger derivation* and *contrastive contamination-based retraining*. To maximize the contamination ability of the poisoned samples, Stage 1 derives a trigger for a given target model. In Stage 2, BOBA retrains the feature extraction layers to reduce the distance between contaminated features. Meanwhile, BOBA trains the classification layers to classify contaminated features into the attack target category. In this way, BOBA injects a batch-oriented backdoor into the model.

### 3.4 TRIGGER DERIVATION

According to the principle of BN layers, adversaries can design a poisoned input whose features are outliers relative to those of other samples in the same batch, thereby increasing the contamination of the poisoned samples. However, since the structure and parameters of each model are specific, it is unrealistic to find a poisoned sample that generalizes across all models. Therefore, for the target model, Stage 1 of our approach aims to derive a specialized trigger that fully exerts its effect in accordance with the model characteristics. Specifically, for each batch $\mathcal{B}_n = (X_n, Y_n)$ sampled from the training dataset $\mathcal{D}$, we randomly select a certain proportion of indexes to obtain $P$ and generate $X_n^p$ according to $\mathcal{T}(X_n, P, \tau)$. Note that the injection poisoning ratio $\eta_i = |P|/n$. In this way, BOBA derives the trigger $\tau$ for the target model $F$ with fixed parameters $\theta$ as follows:

$$\tau^* = \arg\max_{\tau} \sum_{i=1}^{n} \mathcal{L}\Big( F(X_n; \theta)[i], F(X_n^P; \theta)[i] \Big), \quad (3)$$

where we use Cross-entropy as the loss function $\mathcal{L}$ to measure the difference between output logits. Equation 3 strives to optimize a trigger $\tau^*$ that can maximize the Cross-entropy of the output logits between the clean and poisoned batches. In this way, when fed into the model $F$, the derived trigger $\tau^*$ can effectively contaminate the entire batch of sample outputs.

### 3.5 CONTRASTIVE CONTAMINATION-BASED RETRAINING

For the given trained model, we can split it into two parts, i.e., feature extraction layers and classification layers. Since the purpose of our approach is to control the classification results for contaminated samples, the feature extraction layer should aggregate the features of contaminated samples and separate them from those of uncontaminated samples in the feature space. Meanwhile, the classification layers should classify the clustered features of the contaminated samples into the attack target

category $y_t$. Therefore, we adopt contrastive learning to retrain the model. Specifically, since our approach is based on the vulnerability of BN layers, we divide the model $F$ into feature extraction layers $g$ and classification layers $f$ from the last BN layer. In this way, we have $F = f(g(; \theta_1); \theta_2)$, where $\theta_1$ and $\theta_2$ are parameters of $g$ and $f$, respectively.

To train the feature extraction layers, we create positive and negative sample pairs for each sample using the contrastive learning principle. Suppose that we have a batch of clean samples, $X_n$, and its poisoned variant, $X_n^P$. For each sample in $X_n^p$, its positive samples are all the samples in this batch and the samples in $X_n$ with category $y_t$, while its negative samples are all the other samples in $X_n$. For each sample in $X_n$, its positive samples are those with the same category, while its negative samples are all the other samples in $X_n$ and $X_n^p$. In particular, for each sample in $X_n$ with category $y_t$, the samples in $X_n^p$ are its positive samples rather than negative samples. The retraining process aims to reduce the distance between positive sample pairs and increase the distance between negative sample pairs. Therefore, based on the loss function in the existing work (Khosla et al., 2020; Chen et al., 2020), we update the parameters of the feature extraction layers as follows:

$$\theta_1^* = \arg\min_{\theta_1} \frac{1}{n} \sum_{i=1}^{n} \left( -\log \sum \exp(z_i \cdot z_{i+}/t) / \sum \exp(z_i \cdot z_{i-}/t) \right), \qquad (4)$$

where $t$ is the scalar temperature parameter that controls the degree of discrimination of the model for negative samples. The feature representation is calculated by normalizing the output of the feature extraction layers, which is denoted as $\mathcal{Z}_n = \text{normalize}(g(X_n))$. For a sample $x_i$, we denote the feature representations of its positive and negative samples as $z_{i+}$ and $z_{i-}$, respectively.

Meanwhile, to train the classification layers, BOBA fixes the parameters of $g$ and takes the output of $g$ as the input of $f$. In this way, for each batch $\mathcal{B}_n = (X_n, Y_n)$ and a poisoned variant $X_n^p$, we update the classification layer parameters as follows:

$$\theta_2^* = \arg\min_{\theta_2} \left[ \sum_{i=1}^{n} \mathcal{L}(f(g(X_n; \theta_1^*); \theta_2)[i], Y_n[i]) + \sum_{j=1}^{n} \mathcal{L}(f(g(X_n^p; \theta_1^*); \theta_2)[j], y_t) \right]. \qquad (5)$$

Similarly to Equation 3, $\mathcal{L}$ is the Cross-entropy loss function. Equation 5 optimizes the parameters of $f$ to inject a backdoor into the model. For the poisoned batch, the backdoored model $f(; g(\theta_1^*); \theta_2^*)$ classifies all samples as $y_t$. In this way, BOBA can successfully inject a batch-oriented backdoor into the model, revealing the vulnerabilities of the BN layers.

### 3.6 IMPLEMENTATION OF BOBA

Algorithm 1 details the implementation of BOBA. Lines 1-8 describe the trigger derivation process, while Lines 9-17 present the contrastive contamination-based retraining process. Specifically, Line 1 initializes the trigger $\tau$ randomly. In Lines 4-5, we obtain an index set $P$ and poison $X_n$ using the strategy $\mathcal{T}$. Line 6 computes the classification logits for two batches of samples using the model $F$ with fixed parameters $\theta$. Lines 7-8 show the optimization process of the trigger $\tau$ with loss $\mathbf{L}$. Lines 13-14 feed clean and poisoned batches into the feature extraction layers $g$ to obtain the features for these samples. Meanwhile, we normalize these sample features to get the feature representation sets $\mathcal{Z}_n$ and $\mathcal{Z}_n^p$. In Line 15, BOBA updates the parameters of $g$ following Equation 4 using the sets of the normalized feature representations. With the fixed parameters of $g$, Lines 16-17 optimize $\theta_2$ with loss $\mathbf{L}$ to train the classification layers. Finally, Line 18 returns the backdoored model $F^*$.

---

**Algorithm 1** Implementation of BOBA

**Input:** i) $\mathcal{D}$, a training dataset; ii) $F(; \theta) = f(g(; \theta_1); \theta_2)$, a trained deep model with feature extraction layers $g$ and classification layers $f$; iii) $\eta_i$, injection poisoning ratio; iv) $T_1, T_2$, the number of optimization epochs for Stage 1 and 2, respectively; v) $\mathcal{T}$, batch poison strategy.

**Output:** $F^*$, a backdoored model.
1: $\tau \leftarrow$ random initialization
2: **for** $t = 0$ to $T_1$ **do**
3:     **for** each batch $(X_n, Y_n) \subset \mathcal{D}$ **do**
4:         $P \leftarrow \text{sample}([1, 2, ..., n], \eta_i)$
5:         $X_n^p \leftarrow \mathcal{T}(X_n, P, \tau)$
6:         $\hat{Y}_n \leftarrow F(X_n; \theta), \hat{Y}_n^p \leftarrow F(X_n^P; \theta)$
7:         $\mathbf{L} \leftarrow -\sum_{i=1}^{n} \mathcal{L}(\hat{Y}_n[i], \hat{Y}_n^p[i])$
8:         $\tau \leftarrow \text{optimize}(\tau, \mathbf{L})$
9: **for** $t = 0$ to $T_2$ **do**
10:     **for** each batch $(X_n, Y_n) \subset \mathcal{D}$ **do**
11:         $P \leftarrow \text{sample}([1, 2, ..., n], \eta_i)$
12:         $X_n^p \leftarrow \mathcal{T}(X_n, P, \tau^*)$
13:         $\mathcal{Z}_n \leftarrow \text{normalize}(g(X_n))$
14:         $\mathcal{Z}_n^p \leftarrow \text{normalize}(g(X_n^p))$
15:         $\theta_1^* \leftarrow$ following Equation 4$(\mathcal{Z}_n, \mathcal{Z}_n^p)$
16:         $\mathbf{L} \leftarrow \sum_{i=1}^{n} (\mathcal{L}(\hat{Y}_n[i], Y_n[i]) + \mathcal{L}(\hat{Y}_n^p[i], y_t))$
17:         $\theta_2 \leftarrow \text{optimize}(\theta_2, \mathbf{L})$
18: **return** $F^* \leftarrow f(g(; \theta_1^*); \theta_2^*)$

---

## 3.7 THEORETICAL ANALYSIS

To elucidate the core mechanisms that enable a few poisoned samples to control the classification results of an entire batch, we provide a rigorous theoretical foundation for the BOBA attack.

**Theorem 3.1.** *Let $\mathcal{B}_n = (X_n, Y_n)$ be a batch of clean samples. For any sample $\mathbf{x} \in X_n$, the derived trigger $\tau^*$ causes a feature representation $\mathbf{z}^p = g(\mathbf{x} \oplus \tau^*; \theta_1)$ to be an outlier in the batch feature representations $\mathcal{Z}$, which can be characterized by Mahalanobis distance:*

$$D_M(\mathbf{z}^p, \mathcal{Z}) = \sqrt{(\mathbf{z}^p - \boldsymbol{\mu}_{\mathcal{Z}})^\top \boldsymbol{\Sigma}_{\mathcal{Z}}^{-1} (\mathbf{z}^p - \boldsymbol{\mu}_{\mathcal{Z}})} \gg 0, \tag{6}$$

*where $\boldsymbol{\mu}_{\mathcal{Z}}$ is the mean vector of $\mathcal{Z}$, $\boldsymbol{\Sigma}_{\mathcal{Z}}$ is the covariance matrix of $\mathcal{Z}$. The optimized trigger $\tau^*$ ensures $D_M(\mathbf{z}^p, \mathcal{Z}) \gg 0$, making $\mathbf{z}^p$ a statistically extreme outlier that can maximize the distributional shift between the clean and poisoned batches in the feature space.*

**Theorem 3.2.** *Consider a batch of feature representations $\mathcal{Z}^P = \{\mathbf{z}_1, \mathbf{z}_2, ..., \mathbf{z}_{n-1}, \mathbf{z}^p\}$, where $\mathbf{z}^p$ is an outlier according to Theorem 3.1. The mean $\boldsymbol{\mu}_{\mathcal{Z}^P}$ and variance $\boldsymbol{\sigma}_{\mathcal{Z}^P}^2$ of this batch are:*

$$\boldsymbol{\mu}_{\mathcal{Z}^P} = \frac{1}{n} \left( \sum_{i=1}^{n-1} \mathbf{z}_i + \mathbf{z}_p \right) = \boldsymbol{\mu}_{\mathcal{Z}} + \frac{1}{n} \boldsymbol{\delta}, \quad \boldsymbol{\sigma}_{\mathcal{Z}^P}^2 = \frac{1}{n} \sum_{i=1}^{n} (\mathbf{z}_i - \boldsymbol{\mu}_{\mathcal{Z}^P})^2, \tag{7}$$

*where $\boldsymbol{\delta} = \mathbf{z}^p - \boldsymbol{\mu}_{\mathcal{Z}}$ is the deviation of the outlier $\mathbf{z}^p$. In this way, we have:*

$$\begin{aligned}
\boldsymbol{\sigma}_{\mathcal{Z}^P}^2 &= \frac{1}{n} \left[ \sum_{i=1}^{n-1} \left( \mathbf{z}_i - \boldsymbol{\mu}_{\mathcal{Z}} - \frac{\boldsymbol{\delta}}{n} \right)^2 + \left( \mathbf{z}^p - \boldsymbol{\mu}_{\mathcal{Z}} - \frac{\boldsymbol{\delta}}{n} \right)^2 \right] \\
&\approx \frac{1}{n} \left[ \sum_{i=1}^{n-1} (\mathbf{z}_i - \boldsymbol{\mu}_{\mathcal{Z}})^2 + \left( \boldsymbol{\delta} - \frac{\boldsymbol{\delta}}{n} \right)^2 \right] \approx \boldsymbol{\sigma}_{\mathcal{Z}}^2 + \frac{1}{n} \left( \frac{n-1}{n} \right)^2 \boldsymbol{\delta}^2.
\end{aligned} \tag{8}$$

*Therefore, the variance perturbation $\Delta\boldsymbol{\sigma}^2 = \boldsymbol{\sigma}_{\mathcal{Z}^P}^2 - \boldsymbol{\sigma}_{\mathcal{Z}}^2$ converges to a non-zero constant. Since the BN layer normalizes each feature representation using contaminated statistics following equation $\hat{\mathbf{z}}_i = (\mathbf{z}_i - \boldsymbol{\mu}_{\mathcal{Z}}^P)/\sqrt{\boldsymbol{\sigma}_{\mathcal{Z}^P}^2 + \epsilon}$, $\boldsymbol{\sigma}_{\mathcal{Z}^P}^2$ shifts the norm of the normalized clean features $\hat{\mathbf{z}}_i$ as:*

$$\|\hat{\mathbf{z}}_i\| \propto \frac{1}{\|\boldsymbol{\sigma}_{\mathcal{B}^P}\|} = \frac{1}{\sqrt{\|\boldsymbol{\sigma}_{\mathcal{Z}}^2 + \Delta\boldsymbol{\sigma}^2\|}}, \tag{9}$$

*which induces a deterministic shift of the feature representations.*

Theorem 3.1 means that the derived trigger from Stage 1 of BOBA causes an outlier in feature space. According to Theorem 3.2, the outlier leads to a non-zero perturbation of the batch variance, enabling the deterministic shift of contaminated features. Leveraging the deterministic shift, Stage 2 of BOBA injects the batch-oriented backdoors into models to attack batch processing. Specifically, the feature extractor $g$ is optimized to cluster these contaminated features, while the classifier $f$ is trained to map the entire cluster to the target category $y_t$. In this way, BOBA can leverage a few poisoned samples to manipulate the classification results of an entire batch.

## 4 EXPERIMENTS

To evaluate the effectiveness of our approach, we implemented it on top of PyTorch (version 1.13.0). All experiments were conducted on an Ubuntu workstation with one NVIDIA GeForce RTX4090 GPU, one Intel i7-13700K CPU, and 64GB of memory. By default, we set the trigger to a 3×3 square located in the upper right corner of the sample. We used the Adam optimizer with a learning rate $\alpha = 0.01$. We considered scenarios with extremely large batch sizes, where the BN layers of the models directly use batch statistics for more efficient training and inference. In this way, we set all hyperparameters "track_running_stats" of the BN layers to "False" in the following experiments. Meanwhile, we set the training batch size to 1024 and investigated different inference batch sizes, i.e., $n$=512,1024 and 2048.

### 4.1 EXPERIMENTAL SETUP

**Dataset and Model Settings.** We investigated five classical datasets (i.e., MNIST (LeCun et al., 1995), CIFAR-10 (Krizhevsky et al., 2009), GTSRB (Stallkamp et al., 2012), Tiny-ImageNet (Le & Yang, 2015), and ImageNette (Russakovsky et al., 2015)). ImageNette is a subset of the ImageNet

dataset with 10 categories. Since BOBA exploits the vulnerability in BN layers to attack batch processing, we investigated four DL models with BN layers (i.e., CNN_bn, PreAct-ResNet18 (He et al., 2016), VGG19_bn (Simonyan & Zisserman, 2014), and EfficientNet-B3 (Tan & Le, 2019)). CNN_bn is a self-defined structure model that consists of two convolutional layers and two fully connected layers, with a BN layer after each convolutional layer.

**Evaluation Metrics.** To objectively evaluate the performance of the proposed attack approach, we adopted two metrics, i.e., Clean Accuracy (CA) and Attack Contamination Rate (ACR). Specifically, CA denotes the inference accuracy of clean samples, representing the stealthiness of the attack. A higher CA indicates that the attack method has less impact on the normal classification task, suggesting that the performance difference between the poisoned and benign models is minimal. ACR is a novel metric designed to measure the effectiveness of batch-oriented backdoor attacks, indicating the rate at which samples are classified into the target category when poisoned samples are present in the same batch. Note that BOBA aims to alter the classification results of the entire batch, requiring only a very small proportion of samples to be poisoned, which differs significantly from the purpose of traditional backdoors. Therefore, we cannot use a typical ASR as a metric, nor can we make a fair comparison of the effectiveness of BOBA with that of traditional backdoor attacks.

**Poisoning Ratios.** We considered two types of poisoning rates, i.e., the injection poisoning ratio, denoted $\eta_i$, and the attack poisoning ratio, denoted $\eta_a$. Specifically, $\eta_i$ indicates the ratio of poisoned samples in a training batch. The adversary poisons the $\eta_i$ values of the samples in each batch and uses BOBA to inject backdoors. $\eta_a$ indicates the ratio of poisoned samples in an inference batch. The adversary can poison $\eta_a$ samples in a batch to activate the backdoor, thereby altering the classification results of all samples. By default, we set $\eta_i$ to 10% and $\eta_a$ to 1%.

Table 2: Attack performance with different batch sizes.

| Dataset | Model | | $n = 512$ | | | $n = 1024$ | | | $n = 2048$ | |
|---|---|---|---|---|---|---|---|---|---|---|
| | | Benign | BOBA | | Benign | BOBA | | Benign | BOBA | |
| | | CA (%) | CA (%) | ACR (%) | CA (%) | CA (%) | ACR (%) | CA (%) | CA (%) | ACR (%) |
| MNIST | CNN_bn | 99.91 | 99.83 | 80.53 | 99.85 | 99.82 | 78.31 | 98.26 | 98.30 | 77.64 |
| CIFAR-10 | PreAct-ResNet18 | 92.51 | 90.22 | 82.16 | 91.16 | 90.31 | 82.05 | 88.37 | 88.46 | 81.90 |
| | VGG19_bn | 91.35 | 87.39 | 81.43 | 90.43 | 87.28 | 80.39 | 86.42 | 86.96 | 79.34 |
| | EfficientNet-B3 | 66.48 | 61.92 | 79.17 | 64.69 | 59.46 | 79.22 | 61.62 | 59.12 | 80.16 |
| GTSRB | PreAct-ResNet18 | 98.22 | 96.21 | 87.72 | 97.25 | 95.92 | 88.02 | 96.65 | 95.54 | 87.42 |
| | VGG19_bn | 96.17 | 93.77 | 86.26 | 96.19 | 93.38 | 86.13 | 94.37 | 93.06 | 84.33 |
| | EfficientNet-B3 | 86.43 | 84.06 | 81.05 | 85.45 | 83.53 | 81.21 | 84.03 | 83.67 | 78.35 |
| T-ImgNet | PreAct-ResNet18 | 55.96 | 51.35 | 77.73 | 54.17 | 51.62 | 74.31 | 53.26 | 50.96 | 74.79 |
| | VGG19_bn | 47.31 | 45.28 | 78.22 | 46.33 | 45.31 | 77.56 | 45.78 | 44.11 | 77.42 |
| | EfficientNet-B3 | 41.56 | 37.16 | 71.19 | 40.35 | 36.46 | 70.36 | 38.66 | 35.30 | 70.51 |
| ImageNette | PreAct-ResNet18 | 91.92 | 90.08 | 84.12 | 91.03 | 90.10 | 84.21 | 91.03 | 91.13 | 83.47 |
| | VGG19_bn | 82.43 | 80.11 | 85.33 | 81.55 | 79.94 | 83.64 | 81.55 | 80.06 | 82.13 |
| | EfficientNet-B3 | 65.75 | 62.43 | 81.29 | 65.17 | 62.12 | 80.06 | 65.17 | 62.21 | 79.33 |

## 4.2 EXPERIMENTAL RESULTS

Table 2 presents the attack performance of BOBA on four datasets across different model architectures and batch sizes. Specifically, for each case, we trained a benign model and set it as the target model to inject backdoors using our proposed BOBA. To evaluate the stealthiness of BOBA, we compared CA across benign and backdoored models. From the table, we can find that BOBA has CA comparable to that of the benign model. Note that the CA of benign models decreases as $n$ increases, due to the lower generalization ability resulting from a more accurate gradient estimation in larger batch training. Meanwhile, as $n$ increases, the CA of backdoored models becomes more similar to that of benign models, indicating that BOBA even enhances the generalization ability of the models. The experimental results demonstrate that BOBA exhibits satisfactory stealthiness in CA, particularly with larger batch sizes. To evaluate the effectiveness of the BOBA attack, we compared the ACR of backdoored models with different batch sizes. From the table, we find that BOBA can still attack successfully even with extremely large batch sizes (i.e., $n = 2048$), altering the classification results for the majority of batch samples. Taking the CIFAR-10 dataset using the PreAct-ResNet architecture as an example, 1677 out of 2048 samples were misclassified into the specified target category, with only 20 poisoned samples used. Additionally, BOBA demonstrates satisfactory attack performance across various model architectures, suggesting its applicability to BN-based models.

## 4.3 ABLATION STUDIES

We investigated the impact of different stages, trained models, trigger shapes, trigger positions, and poisoning ratios on BOBA performance. Due to space limitations, we only present the first two experimental results in this section. Please refer to the appendix for more ablation studies.

**Impact of Different Stages.** The implementation of BOBA consists of two stages, where the first aims to derive triggers from trained models, and the second to retrain the models using those triggers. To evaluate the effectiveness of the derived triggers, we considered two variants of BOBA (i.e., "Fixed" and "Random"), in which we retrained the models in Stage 2 with fixed or random triggers, respectively, rather than the triggers derived in Stage 1.

Specifically, fixed triggers are the $3\times3$ squares with a white color (i.e., $R, G, B$=255), and random triggers are sampled from a Gaussian distribution. Meanwhile, to examine the impacts of these two stages, we considered another variant "Stage 1", which directly uses the triggers derived in Stage 1 to attack the models without implementing Stage 2. Table 3

Table 3: Ablation study on two stages.

| Dataset | Fixed (%) | | Random (%) | | Stage 1 (%) | | BOBA (%) | |
|---|---|---|---|---|---|---|---|---|
| | CA | ACR | CA | ACR | CA | ACR | CA | ACR |
| MNIST | 99.15 | 10.65 | 99.24 | 5.64 | 99.90 | 2.07 | 99.82 | 78.31 |
| CIFAR-10 | 90.73 | 6.76 | 91.31 | 2.56 | 91.13 | 2.21 | 90.31 | 82.05 |
| GTSRB | 94.33 | 8.15 | 94.25 | 6.82 | 96.42 | 4.41 | 95.92 | 88.02 |
| T-ImgNet | 50.26 | 1.97 | 50.56 | 3.12 | 53.24 | 0.16 | 51.62 | 74.31 |
| ImageNette | 87.92 | 6.24 | 88.27 | 3.31 | 91.23 | 3.38 | 90.10 | 84.21 |

presents the results of the ablation study in two stages when the batch size $n$=1024. Compared with "Fixed" and "Random", BOBA achieves significantly better attack performance. This means that triggers derived from a model can help inject backdoors more effectively, demonstrating the effectiveness and necessity of Stage 1. Similarly, based on the results of "Stage 1" and "BOBA", we can observe that Stage 2 significantly improves the effectiveness of attacks, demonstrating the necessity of Stage 2.

**Impact of Trained Models.** For a trained model, BOBA generates triggers using our proposed trigger derivation mechanism in Stage 1, then uses poisoned samples embedded with those triggers to retrain the model in Stage 2. However, BOBA may fail when applied to an untrained model. This is because untrained models lack sufficient knowledge to derive effective triggers. Meanwhile, since the generated triggers are ineffective, our contrastive contamination-based retraining may also fail to establish a connection between the triggers and the attack target category. To evaluate the impact of trained models on the attack performance of BOBA, we considered a variant of BOBA (i.e., "Untrained") that derives the trigger for an untrained model. Then, "Untrained" trains

Table 4: Ablation study on untrained models.

| Dataset | Bengin (%) | Untrained (%) | | BOBA (%) | |
|---|---|---|---|---|---|
| | CA | CA | ACR | CA | ACR |
| MNIST | 99.85 | 98.52 | 4.31 | 99.82 | 78.31 |
| CIFAR-10 | 91.16 | 61.33 | 7.54 | 90.30 | 82.05 |
| GTSRB | 97.25 | 65.42 | 4.30 | 95.92 | 88.02 |
| T-ImgNet | 54.17 | 30.18 | 1.17 | 51.62 | 74.31 |
| ImageNette | 91.03 | 68.24 | 8.33 | 90.10 | 84.21 |

the model and implements Stage 2 of BOBA. Table 4 presents the results of the ablation study when the batch size $n$=1024. The table shows that the CA and ACR of "Untrained" are lower than those of BOBA, indicating that obtaining triggers for untrained models is difficult and that both models and triggers are hard to converge, resulting in the failure of the attack.

## 5 DISCUSSION

To investigate possible solutions to the vulnerability revealed in the BN layers, we explored adaptive defenses against BOBA, discussed the resistance of BOBA to existing defenses (i.e., STRIP (Gao et al., 2019), Neural Cleanse (Wang et al., 2019), and SentiNet (Chou et al., 2020)), and analyzed the limitations and future work. Due to space limitations, this section only presents the investigation of adaptive defenses. Please refer to the appendix for more discussion.

### 5.1 ADAPTIVE DEFENSES

In response to the security risks that we revealed in the BN layers, we discuss possible effective adaptive defenses against our proposed BOBA. Adaptive defenses aim to reduce the effectiveness of our BOBA while maintaining model availability. Assuming the defenders have prior knowledge of BOBA, they can leverage its characteristics to design adaptive defenses. We considered two

defense scenarios, i.e., black-box and white-box scenarios. In black-box scenarios where defenders are unable to access or alter the model parameters, we consider two sample-level adaptive defenses, i.e., sample noise addition and statistical variance check. In white-box scenarios, assuming that defenders can modify the parameters of the backdoored model to defend against BOBA, we consider two model-level adaptive defenses, i.e., differential privacy and hyperparameter repair.

Table 5: Performance of sample noise addition against BOBA.

| $\sigma$ | MNIST (%) | | | CIFAR-10 (%) | | | GTSRB (%) | | | T-ImgNet (%) | | | ImageNette (%) | | |
|---|---|---|---|---|---|---|---|---|---|---|---|---|---|---|---|
| | Benign | BOBA | | Benign | BOBA | | Benign | BOBA | | Benign | BOBA | | Benign | BOBA | |
| | CA | CA | ACR | CA | CA | ACR | CA | CA | ACR | CA | CA | ACR | CA | CA | ACR |
| 0 | 99.85 | 99.82 | 78.31 | 91.16 | 90.31 | 82.05 | 97.25 | 95.92 | 88.02 | 54.17 | 51.62 | 74.31 | 91.03 | 90.10 | 84.21 |
| 0.05 | 98.02 | 98.42 | 74.52 | 83.21 | 81.27 | 70.25 | 90.36 | 88.24 | 73.48 | 48.99 | 45.24 | 68.27 | 80.51 | 79.43 | 72.82 |
| 0.10 | 94.17 | 93.56 | 65.13 | 68.57 | 65.13 | 61.30 | 83.49 | 77.17 | 65.56 | 36.12 | 34.30 | 52.60 | 70.12 | 66.40 | 63.35 |
| 0.20 | 90.06 | 88.27 | 41.24 | 53.22 | 48.50 | 36.12 | 64.55 | 60.36 | 42.93 | 30.89 | 22.86 | 33.44 | 51.41 | 49.71 | 41.47 |
| 0.30 | 82.51 | 74.33 | 18.33 | 42.17 | 37.61 | 25.33 | 48.12 | 42.35 | 18.64 | 21.52 | 15.10 | 16.37 | 42.82 | 35.25 | 28.47 |

**Sample Noise Addition.** Since BOBA can only be activated by optimized triggers on poisoned samples, defenders can counter attacks by adding noise to each sample, thus destroying potential triggers. To investigate the performance of the sample noise addition defense, we conducted experiments on four datasets. Specifically, we added random Gaussian noise $z$ to each input sample $x$ following equation $x_m = (1 - \sigma)x + \sigma z$ ($z \sim \mathcal{N}(0, \mathbf{I})$, where $\sigma$ controls the noise intensities. We considered different noise intensities, i.e., $\sigma = 0$, 0.05, 0.10, 0.20, and 0.30, respectively, when the batch size $n$=1024. The experimental results are presented in Table 5, which shows that adding low-intensity noise does not reduce the effectiveness of BOBA. In contrast, the addition of high-intensity noise results in a significant decrease in CA, rendering the classification task unusable. Therefore, the sample noise addition defense is not a viable adaptive defense against BOBA.

**Statistical Variance Check.** Since BOBA aims to contaminate the entire batch using a few poisoned samples, clean samples should make up the majority of the batch. Therefore, defenders can identify and exclude poisoned samples by counting the overall statistics of the batch to perform a Statistical Variance Check (SVC). To investigate the performance of the SVC defense, we conducted experiments considering three different batch sizes, i.e., $n$=512, 1024, and 2048, on four datasets, respectively. Specifically, for a batch of $n$ samples with a 1% ratio of poisoned samples, we calculate the mean and variance of the pixel values across the batch, denoted by $\mu_B$ and $\sigma_B$, respectively. In this way, we can identify that a sample $x$ is poisoned if $|\mu - \mu_B| > \eta \cdot \sigma_B$, where $\mu$ is the mean value of the pixels of $x$. For each case, we changed the degree of $\eta$ until the False Positive Rate (FPR) was

Table 6: Defense performance of SVC.

| Dataset | F1 Score | | |
|---|---|---|---|
| | $n$=512 | $n$=1024 | $n$=2048 |
| MNIST | 0.15 | 0.14 | 0.15 |
| CIFAR-10 | 0.08 | 0.09 | 0.14 |
| GTSRB | 0.10 | 0.09 | 0.10 |
| T-ImgNet | 0.15 | 0.14 | 0.13 |
| ImageNette | 0.13 | 0.12 | 0.11 |

10% and calculated the F1 score. The experimental results are presented in Table 6, which indicates that the statistical variance check defense is ineffective in identifying poisoned samples, suggesting that this defense is not a viable adaptive defense against BOBA.

**Differential Privacy.** Since the BOBA injects the backdoor by retraining model parameters, the attack may lose its effectiveness when the parameters are perturbed. Therefore, defenders can compromise the injected backdoor by imposing Differential Privacy (DP) (Arachchige et al., 2019) on the model parameters. To investigate the performance of differential privacy defense, we conducted experiments considering various levels of DP when the batch size $n$=1024. Specifically, we use $\varepsilon_r$ to specify the privacy budget of DP, where a lower value of $\varepsilon_r$ indicates a higher level of DP. Note that when $\varepsilon_r$=$+\infty$, DP is not applied to the models.

Table 7: Performance of differential privacy.

| $\varepsilon_r$ | MNIST (%) | | CIFAR-10 (%) | | GTSRB (%) | | T-ImgNet (%) | | ImageNette (%) | |
|---|---|---|---|---|---|---|---|---|---|---|
| | CA | ACR | CA | ACR | CA | ACR | CA | ACR | CA | ACR |
| $+\infty$ | 99.82 | 78.31 | 90.31 | 82.05 | 95.92 | 88.02 | 51.62 | 74.31 | 90.10 | 84.21 |
| 30 | 92.11 | 75.34 | 83.71 | 68.21 | 92.16 | 71.36 | 42.22 | 59.05 | 85.84 | 71.20 |
| 20 | 90.24 | 71.17 | 74.50 | 56.93 | 85.23 | 65.50 | 36.41 | 42.61 | 72.17 | 55.22 |
| 10 | 85.06 | 59.28 | 65.42 | 39.88 | 81.10 | 54.83 | 29.69 | 35.78 | 58.36 | 40.62 |
| 5 | 61.30 | 15.31 | 40.06 | 17.25 | 62.98 | 38.22 | 12.66 | 18.42 | 42.25 | 19.34 |

The experimental results are presented in Table 7, which shows that when the DP level is low, it cannot destroy the backdoors injected into the modes, resulting in defense failures. In contrast, when

DP is high, there is a significant loss of CA, indicating that the models are unusable. Therefore, differential privacy defense is not a viable adaptive defense against BOBA.

**Hyperparameter Repair.** Since the BN vulnerability can be exploited by BOBA only when the "track_running_stats" hyperparameter in the last BN layer of a model is set to "False", defenders can change this hyperparameter to "True" to defend against attacks. Meanwhile, considering the differences in the BN parameters under different "track_running_stats" settings, shown in Table 1, to make models available, the defenders must assign values to the additional parameters, i.e., "running_mean" and "running_var".

To effectively perform hyperparameter repair defense, we considered two assignment strategies, i.e., random initialization and Gaussian distribution. The former strategy randomly initializes the values of two parameters. The latter strategy follows the Gaussian distribution to assign values, i.e., running_mean= 0 and running_var= 1. We conducted experiments to investigate the performance of hyperparameter repair defense across four datasets using various model architectures, with a batch size of $n$=1024. The experimental results show that when "track_running_stats" is set to "True", BOBA fails to attack in any case, indicating the success of defenses. How-

Table 8: Performance of hyperparameter repair.

| Dataset | Model | Clean Accuracy (%) | | |
|---|---|---|---|---|
| | | No Defense | Random | Gaussian |
| MNIST | CNN_bn | 99.82 | 73.18 | 91.33 |
| CIFAR-10 | PreAct-ResNet18 | 90.31 | 61.42 | 79.78 |
| | VGG19_bn | 87.28 | 49.50 | 74.12 |
| | EfficientNet-B3 | 59.46 | 40.12 | 51.97 |
| GTSRB | PreAct-ResNet18 | 95.92 | 71.88 | 87.25 |
| | VGG19_bn | 93.38 | 67.60 | 82.69 |
| | EfficientNet-B3 | 83.53 | 57.51 | 74.30 |
| T-ImgNet | PreAct-ResNet18 | 51.62 | 27.24 | 41.06 |
| | VGG19_bn | 45.31 | 21.83 | 32.67 |
| | EfficientNet-B3 | 36.46 | 15.16 | 25.47 |
| ImageNette | PreAct-ResNet18 | 90.10 | 58.56 | 74.33 |
| | VGG19_bn | 79.94 | 42.35 | 63.78 |
| | EfficientNet-B3 | 62.12 | 36.12 | 42.51 |

ever, both assignment strategies lead to a decrease in Clean Accuracy (CA), shown in Table 8. From the table, we can find that, when using the Gaussian distribution strategy, the drop in accuracy is acceptable. Therefore, the use of hyperparameter repair defense is regarded as a viable defense method, despite its drawbacks of operational complexity and accuracy loss.

## 5.2 LIMITATION AND FUTURE WORK

In the discussion above, we conclude that when defenders identify security risks in the BN layers and focus on examining the hyperparameter "track_running_stats", they can implement a hyperparameter repair defense to defend against BOBA, albeit at some accuracy loss. In this way, we can investigate more effective and available batch-oriented backdoor attacks that can bypass such adaptive defenses. Meanwhile, our work does not focus on the visual stealthiness of triggers. Therefore, more advanced batch-oriented attacks with invisible triggers warrant further study. Additionally, whether the other normalization layers (e.g., LN, IN, and GN) in models can be exploited to inject backdoors is also a worthy area for future study.

## 6 CONCLUSION

This paper reveals a potential threat to DL models posed by their BN layers when processing extremely large batch sizes. To demonstrate the dangers posed by security threats, we leveraged the properties of BN layers to design a novel batch-oriented backdoor attack, BOBA. Based on our proposed trigger derivation mechanism and contrastive contamination-based retraining method, BOBA can control the classification results of all samples in a batch by poisoning only a few of them. Comprehensive experiments across various well-known datasets and models demonstrate the effectiveness of BOBA, highlighting the detrimental impact of security risks in BN layers.

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

# A APPENDIX

## A.1 ADDITIONAL ABLATION STUDIES

**Impact of Trigger Shapes.** The first stage of our approach (i.e., trigger derivation) generates specific triggers for a trained model. The shape of these triggers significantly impacts the effectiveness of the optimization process. To evaluate the impact of different trigger shapes, we considered various trigger shapes and conducted experiments across different datasets. Figure 4 presents examples of poisoned samples embedded with different trigger shapes of CIFAR-10. Table 9 shows the results of the ablation study on different trigger shapes. From this table, we can observe that as the number of trigger pixels increases, the ACR of BOBA increases. This means that more pixels can contain more information, allowing BOBA to derive the best triggers to maximize contamination on clean samples. Meanwhile, we observe that as the ACR of the attack increases, the CA decreases, indicating that injecting backdoors consumes part of the classification capacity of models, resulting in a decrease in classification performance. Therefore, when the capacity of the model reaches its upper bound, it is difficult to improve the ACR while maintaining the CA by increasing the trigger size.

Table 9: Ablation study on different trigger shapes.

| Trigger Shape | MNIST (%) | | CIFAR-10 (%) | | GTSRB (%) | | T-ImgNet (%) | |
|---|---|---|---|---|---|---|---|---|
| | CA | ACR | CA | ACR | CA | ACR | CA | ACR |
| 1×1 | 99.90 | 1.39 | 83.25 | 3.30 | 96.42 | 5.35 | 53.23 | 0.32 |
| 1×2 | 99.91 | 15.54 | 82.74 | 15.96 | 96.27 | 13.48 | 52.42 | 12.58 |
| 1×3 | 99.89 | 38.38 | 81.63 | 41.52 | 95.96 | 25.47 | 52.10 | 43.35 |
| 3×3 | 99.82 | 78.31 | 90.31 | 82.05 | 95.92 | 88.02 | 51.62 | 74.31 |
| 5×5 | 99.61 | 81.17 | 77.68 | 83.34 | 88.61 | 88.24 | 43.41 | 75.26 |
| 7×7 | 99.42 | 81.94 | 77.34 | 84.20 | 87.14 | 89.97 | 41.40 | 75.55 |

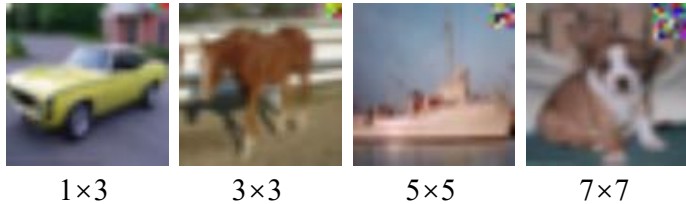

| 1×3 | 3×3 | 5×5 | 7×7 |

Figure 4: Samples with triggers of different shapes.

**Impact of Trigger Positions.** To investigate the impact of trigger positions on BOBA, we conducted experiments involving three different trigger positions. In Table 10, we use the notation "Corner", "Edge", and "Center" to represent the three typical cases, where triggers are placed in the corners, middle of edges, and center of all samples, respectively. Note that, for "Corner" and "Edge", we considered the average attack performance of triggers placed in all four corners and edges of the samples, respectively. From this table, we find that BOBA shows similar attack performance across triggers with different positions. Since the triggers in "Corner" are much stealthier, we have embedded them in the upper right corners of the samples by default.

Table 10: Ablation study on different trigger positions.

| Trigger Positon | MNIST (%) | | CIFAR-10 (%) | | GTSRB (%) | | T-ImgNet (%) | |
|---|---|---|---|---|---|---|---|---|
| | CA | ACR | CA | ACR | CA | ACR | CA | ACR |
| Corner | 99.82 | 78.31 | 90.31 | 82.05 | 95.92 | 88.02 | 51.62 | 74.31 |
| Edge | 99.71 | 81.84 | 88.14 | 82.57 | 94.25 | 88.59 | 50.02 | 74.48 |
| Center | 99.60 | 80.96 | 87.35 | 86.13 | 94.37 | 89.10 | 49.93 | 74.35 |

**Impact of Injection Poisoning Ratios.** In our approach, we use $\eta_i$ to control the ratios of poisoned samples in batches during the training phase. To explore the impact of $\eta_i$ on the performance of our approach, we trained backdoored models with different $\eta_i$ on various datasets. Table 11 shows the results of the ablation study on different $\eta_i$. From this table, we find that as the values of $\eta_i$ increase, the CA decreases while the ACR values increase. When $\eta_i$ increases from 10% to 20%, the

corresponding ACR keeps increasing with gradually lower growth rates. Meanwhile, the CA of the backdoored models continues to decrease, indicating that too many poisoned samples during training do not improve the performance of our proposed attack.

Table 11: Ablation study on different $\eta_i$.

| $\eta_i(\%)$ | MNIST (%) | | CIFAR-10 (%) | | GTSRB (%) | | T-ImgNet (%) | |
|---|---|---|---|---|---|---|---|---|
| | CA | ACR | CA | ACR | CA | ACR | CA | ACR |
| 1 | 99.90 | 15.86 | 92.37 | 18.21 | 95.93 | 15.71 | 54.56 | 5.82 |
| 2 | 99.91 | 41.52 | 92.05 | 31.18 | 95.82 | 35.43 | 53.14 | 16.87 |
| 5 | 99.85 | 68.47 | 91.41 | 45.23 | 95.42 | 67.82 | 51.90 | 36.22 |
| 10 | 99.82 | 78.31 | 90.31 | 82.05 | 95.92 | 88.02 | 51.62 | 74.31 |
| 20 | 98.51 | 81.43 | 85.35 | 83.02 | 94.30 | 88.22 | 42.37 | 75.64 |

**Impact of Attack Poisoning Ratios.** In our approach, we use $\eta_a$ to control the ratios of poisoned samples within batches during inference. To explore the impact of $\eta_a$ on the performance of BOBA, we evaluated BOBA across different $\eta_a$ values on various datasets. Table 12 shows the results of the ablation study on different $\eta_a$. From this table, we observe that as the values of $\eta_a$ increase, the ACR values also increase. In this way, as the inference batch size increases, the adversary only needs to increase the number of poisoned samples to maintain high attack effectiveness. Meanwhile, from the table, we can find that when $\eta_a$ increases from $2\%$ to $4\%$, the value of ACR remains stable and almost no longer increases, meaning that it is unnecessary to poison too many samples to obtain satisfactory ACR.

Table 12: Ablation study on different $\eta_a$.

| $\eta_a(\%)$ | ACR (%) | | | |
|---|---|---|---|---|
| | MNIST | CIFAR-10 | GTSRB | T-ImgNet |
| 0.1 | 51.68 | 62.39 | 70.10 | 60.06 |
| 0.5 | 62.35 | 73.11 | 81.23 | 59.54 |
| 1 | 78.31 | 82.05 | 88.02 | 74.31 |
| 2 | 80.54 | 83.40 | 89.92 | 75.33 |
| 4 | 80.79 | 83.66 | 90.03 | 76.05 |

## A.2 ADDITIONAL DISCUSSION

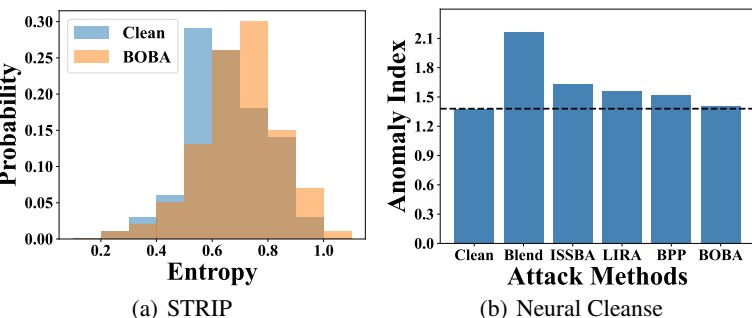

(a) STRIP      (b) Neural Cleanse

Figure 5: The results of STRIP and Neural Cleanse.

**STRIP** (Gao et al., 2019) detects poisoned samples based on the prediction randomness of samples generated by embedding various patterns in suspicious samples, which means that when embedded with different patterns, clean samples will be randomly classified as different categories, while poisoned samples will always be classified as the same category. The entropy of the average prediction of the samples quantifies the randomness. Figure 5 (a) shows the average entropy of clean and poisoned samples of CIFAR-10. From the figure, we find that the entropy of our proposed approach is close to that of the clean model, indicating that STRIP has difficulty detecting our attack.

**Neural Cleanse (NC)** (Wang et al., 2019) aims to invert the different triggers that can change the classification results of clean samples for each category. By detecting whether the inversion trigger (i.e., anomaly index) exceeds the anomaly threshold, NC can determine whether a model is poisoned. Therefore, for a backdoor attack, the smaller the anomaly index of its inversion triggers, the more difficult it is for NC to detect. Figure 5 (b) shows the NC results of our approach and four advanced backdoor methods (i.e., Blend (Chen et al., 2017), ISSBA (Li et al., 2021), LIRA (Doan et al., 2021), and BppAttack (Wang et al., 2022)) on the CIFAR-10 dataset. From this figure, we can find that the anomaly index of our approach is slightly higher than that of a clean model but significantly lower than that of existing attacks. The comparison results reveal that NC cannot invert the effective triggers of our backdoored model, indicating that BOBA can bypass NC detection.

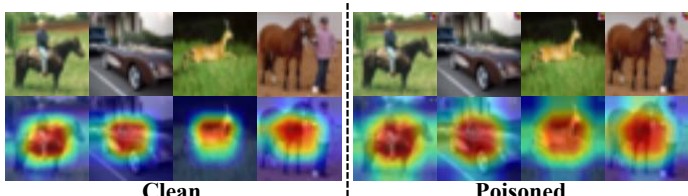

**Clean**  **Poisoned**

Figure 6: Heatmaps of clean and poisoned samples.

**SentiNet** (Chou et al., 2020) leverages Grad-CAM (Selvaraju et al., 2017) to identify the most important regions (i.e., heatmaps) of samples that contribute to the model classification results. By comparing Grad-CAM across different samples, SentiNet can identify the trigger regions in each sample. Figure 6 shows the heatmaps of clean and poisoned samples with a backdoored model trained by our approach. From this figure, we observed no obvious differences between the heatmaps, indicating that BOBA is robust to both SentiNet and other Grad-CAM-based defense methods.

