# OpenReview forum: "One Bad Sample May Spoil the Whole Batch: A Novel Backdoor-Like Attack Towards Large Batch Processing"
_ICLR.cc/2026/Conference — Submitted to ICLR 2026_

### Official Review · Reviewer_D37u · 2025-10-28

**Soundness:** 2
**Presentation:** 3
**Contribution:** 2
**Rating:** 4
**Confidence:** 4

**Summary:**

This paper proposes a novel Batch-Oriented Backdoor Attack named BOBA, which aims to control the classification results of all the samples in a batch by poisoning only one of them. BOBA exploits an intrinsic mechanism of the Batch Normalization (BN) layer in deep learning models, where the BN layer relies on the statistics of the current batch. This allows a single anomalous sample to contaminate the mean and variance of the entire batch, thereby affecting the feature representations of all other normal samples within it. Notably, for CIFAR-10, BOBA can make 848 of 1024 samples within a batch misclassified when manipulating only 10 poisoned samples, indicating the harmfulness of security risks in the BN layers.

**Strengths:**

1. This work reveals a security risk in the Batch Normalization (BN) layer of deep learning models and demonstrates that its intrinsic mechanism can be exploited to implant a backdoor.
2. This backdoor attack considers the scenario of batch data processing, which has a certain degree of practical relevance.

**Weaknesses:**

1. The approach of designing attacks by exploiting the intrinsic mechanisms of deep learning models is not highly novel, as similar research already exists. For example, Yuan et al. [1] designed an attack by utilizing the random neuron dropping mechanism of Dropout, while Wei et al. [2] implanted a backdoor by leveraging the down-sampling mechanism in DL models.
2. The paper's threat model states that the backdoored model is delivered to the user as a black-box product. However, the experimental section severely lacks an evaluation against current, state-of-the-art, general-purpose black-box backdoor defense methods, such as [3] [4] [5]. These defense methods require no prior knowledge of the backdoor attack and align perfectly with the paper's black-box setting, yet the paper lacks comparative experiments for BOBA against these SOTA defenses.
3. In Section 4.1, the paper sets the default training batch size to n = 1024 , while in Table 2, the authors evaluate three different inference batch sizes (512, 1024, 2048). The authors need to clarify the experimental setup: for each column in Table 2 (e.g., (n = 512)), are the results obtained from a model trained with the corresponding batch size (n = 512), or are all results derived from a single model trained with the default batch size (n = 1024)? In other words, did the experiments train a separate model for each inference batch size n , or did they use one fixed model trained with ( n = 1024 ) and test it under different batch sizes? If the training batch size is fixed, how does BOBA perform at much larger batch sizes (e.g., 10,000)? If a different model must be trained for each batch size, this would weaken the practicality of the attack.
4. An analysis of the computational cost of the BOBA training process is missing.
[1] Yuan A, Oprea A, Tan C. Dropout attacks[C]//2024 IEEE Symposium on Security and Privacy (SP). IEEE, 2024: 1255-1269.
[2] Wei C, Lee Y, Chen K, et al. Aliasing backdoor attacks on pre-trained models[C]//32nd USENIX Security Symposium (USENIX Security 23). 2023: 2707-2724.
[3] Guo J, Li Y, Chen X, et al. SCALE-UP: An Efficient Black-box Input-level Backdoor Detection via Analyzing Scaled Prediction Consistency[C]//ICLR. 2023.
[4] Zeng Y, Park W, Mao Z M, et al. Rethinking the backdoor attacks' triggers: A frequency perspective[C]//Proceedings of the IEEE/CVF international conference on computer vision. 2021: 16473-16481.
[5] Yang Y, Jia C, Yan D K, et al. Sampdetox: Black-box backdoor defense via perturbation-based sample detoxification[J]. Advances in Neural Information Processing Systems, 2024, 37: 121236-121264.

**Questions:**

1. What is the total computational time required for the entire training process of BOBA? Compared to training a standard benign model on the same dataset, by what factor does this overhead increase?
2. During the trigger's gradient optimization or the inference process, are the trigger's pixel values constrained to a valid image data range (e.g., [0, 1] or [0, 255])? Because illegal pixel values can often influence a model's output more significantly, the authors need to clarify this setup. If the trigger contains illegal pixel values, the credibility of the reported high attack success rates would be questionable.
3. The trigger optimized in this paper is in the form of a patch. Could it be replaced with a global perturbation, for example, by blending the perturbation with the image at a certain ratio to serve as the trigger? Would the effectiveness of the attack be affected by this setting?
4. The paper's experimental evaluation is primarily focused on low- or mid-resolution image datasets. How does the proposed BOBA attack perform on higher-resolution images (e.g., 224x224)?
5. What is the Attack Success Rate (ASR) of the trigger optimized in Stage 1 of BOBA?

---

> ### Author Response · Authors · 2025-11-26
>
> **AW1:** Sorry for the confusion. Indeed, BOBA exploits the intrinsic mechanisms of deep learning models (i.e., BN layers) to achieve attacks. However, BOBA is totally different from the works in [1] and [2]. Specifically, BOBA is a novel attack paradigm with a fundamentally different purpose. BOBA aims to control the classification of an entire batch of samples, unlike traditional backdoor attacks that target a single sample. Our work is the first to reveal a critical security risk inherent in BN layers in large batch processing scenarios, establishing a new threat beyond the traditional backdoor framework.
>
> **AW2:** Thanks for the suggestion. We conducted new experiments to evaluate BOBA's attack performance against four advanced black-box backdoor defense methods on the CIFAR-10 dataset. Specifically, we considered two detection defenses, i.e., **SCALE-UP [a] and Frequency [b]**, and two purification defenses, i.e., **ZIP [c] and SampDetox [d]**.
> From the following tables, we find that BOBA still achieves high PA and ACR against ZIP and SampDetox, demonstrating its high robustness to purification defenses.
> For SCALE-UP detection, BOBA achieves high FRR and FAR, indicating that our attack confuses the detection. Specifically, SCALE-UP amplifies the pixel values of suspicious samples with different multiples and feeds them together into the models to obtain their predictions. By analyzing the predictions, SCALE-UP can identify poisoned samples. However, since BOBA enables backdoor knowledge transfer among samples within a batch, it disrupts SCALE-UP sample predictions, thereby failing the SCALE-UP defense.
> For Frequency detection, we find that Frequency identifies the poisoned samples from BOBA.  This is because our work does not focus on the visual stealthiness of triggers, which results in the poisoning samples being detected, as discussed in **Section 5.2 (Limitation and Future Work)** of our paper.
>
>
> | Detection | FRR | FAR |
> |:---:|:---:|:---:|
> | SCALE-UP | 27.56 | 45.90 |
> | Frequency | 10.11 | 21.26 |
>
> | Purification | CA (\%) | PA (\%) | ACR (\%) |
> |:---:|:---:|:---:|:---:|
> | ZIP | 81.37 | 21.56 | 65.31 |
> | SampDetox | 85.24 | 41.12 | 50.26 |
>
> **Reference:**\
> [a] Guo et al.SCALE-UP: An Efficient Black-box Input-level Backdoor Detection via Analyzing Scaled Prediction Consistency. ICLR 2023.\
> [b] Zeng et al. Rethinking the backdoor attacks' triggers: A frequency perspective. ICCV 2021.\
> [c] Shi et al. Black-box backdoor defense via zero-shot image purification. NeurIPS 2023.\
> [d] Yang et al. SampDetox: Black-box Backdoor Defense via Perturbation-based Sample Detoxification. NeurIPS 2024.
>
> **AW3:** Sorry for the confusion. We clarify that the experimental results with different inference batch sizes (n=512, 1024,2048) in Table 2 are obtained from a single model trained with the default batch size (n=1024). In other words, the inference batch size of our approach is independent of the training batch size. We emphasized the above experimental setup in Section 4.1 of the revised paper.
>
> Meanwhile, we conducted new experiments over a larger range of batch sizes (64-8192) on the CIFAR-10 dataset. From the following table, we can find that BOBA achieves satisfactory attack performance with different batch sizes, even with a large batch size (e.g., n=8192).
>
> | Batch size | Benign | &emsp; BOBA |
> |:---:|:---:|:---:|
> |  | CA (%) | CA(%)/ACR(%) |
> | 64 | 88.18 | 88.21 / 80.02 |
> | 128 | 91.05 | 90.17 / 82.10 |
> | 256 | 90.33 | 89.33 / 81.43 |
> | 512 | 92.51 | 90.22 / 82.16 |
> | 1024 | 91.16 | 90.31 / 82.05 |
> | 2048 | 88.37 | 88.46 / 81.90 |
> | 4096 | 89.45 | 88.34 / 80.77 |
> | 8192 | 90.50 | 90.31 / 81.20 |
>
> **AW4:** Thanks for the suggestion.
> We conducted new experiments to investigate the time overhead of BOBA on the CIFAR-10 dataset. From the following table, we find that BOBA's overall time overhead is **1.177** times that of training a benign model. This overhead is acceptable for attackers.
>
> | Method | Overhead (h) | Factor (vs. Benign) |
> |:---:|:---:|:---:|
> | Benign | 0.328 | 1.000 |
> | Stage 1 | 0.162 | 0.494 |
> | Stage 2 | 0.224 | 0.683 |
> | BOBA | 0.386 | 1.177 |

---

> > ### Author Response · Authors · 2025-11-26
> >
> > **AQ1:** Thanks for the suggestion. Please refer to **AW4** for our answer.
> >
> > **AQ2:** Thanks for the suggestion. We conducted new experiments to determine whether pixel values in poisoned samples are illegal during inference. Specifically, we analyzed 2700 pixel values from the CIFAR-10 test set, using a 1% attack-poisoning ratio, 3$\times$3 trigger size, and 3 color channels. The experimental results show that only **4.48%** of the pixel values are illegal.  Meanwhile, when we forcibly modify the pixel values to be legal, the ACR of the BOBA attack remains at **82.05%**, demonstrating that whether the pixel values are illegal does not impact the attack performance.
> >
> > **AQ3:** Thanks for the suggestion.
> > We conducted new experiments to investigate the attack performance of BOBA using global perturbations as triggers. Specifically, in Stage 1 of BOBA, we optimize a perturbation trigger of the same size as the image samples and blend it with the samples. From the following table, we can find that when the image size is small, the global perturbation triggers are effective.  However, when the image size is large, it is difficult to optimize a perturbation trigger that can be used to inject batch-oriented backdoors with satisfactory attack performance. Therefore, for the proposed BOBA, optimizing patches rather than global perturbations as triggers is a more reasonable choice.
> >
> > | &emsp; Dataset | CA (%) | ACR (%) |
> > |:---:|:---:|:---:|
> > | MNIST | 99.82 | 78.22 |
> > | CIFAR-10 | 90.25 | 81.94 |
> > | GTSRB | 94.30 | 87.21 |
> > | Tiny-ImageNet | 50.06 | 47.33 |
> > | ImageNette | 81.33 | 11.07 |
> >
> > **AQ4:** Thanks for the suggestion.
> > We conducted new experiments on the **ImageNette** dataset, a subset of ImageNet comprising 10 categories and images of size 224$\times$224. From the following table, we can find that BOBA remains effective on high-resolution images.
> >
> > | &emsp; Model | &emsp; &emsp;&emsp;  $n=512$ | | &emsp; &emsp; &emsp; $n=1024$ | | &emsp;  &emsp; &emsp; $n=2048$ | |
> > | :-------------: | :-----: | :------------: | :------: | :------------: | :------: | :------------: |
> > | | Benign | BOBA | Benign | BOBA | Benign | BOBA |
> > | | CA (%) | CA(%)/ACR(%) | CA (%) | CA(%)/ACR(%) | CA (%) | CA(%)/ACR(%) |
> > | PreAct-ResNet18 | 55.96 | 51.35 / 77.73 | 54.17 | 51.62 / 74.31 | 53.26 | 50.96 / 74.79 |
> > | VGG19\_bn | 47.31 | 45.28 / 78.22 | 46.33 | 45.31 / 77.56 | 45.78 | 44.11 / 77.42 |
> > | EfficientNet-B3 | 41.56 | 37.16 / 71.19 | 40.35 | 36.46 / 70.36 | 38.66 | 35.30 / 70.51 |
> >
> > We added the above experimental results to **Table 2** in the revised paper.
> > Additionally, we added the experimental results from ablation studies and discussions on the ImageNette dataset in **Tables 3, 4, 5, 6, 7, and 8** of the revised paper, which reach the same conclusions as the original paper.
> >
> > **AQ5:** Sorry for the confusion. We believe you are referring to the Attack Contamination Rate (ACR), which is a novel metric designed to measure the effectiveness of batch-oriented backdoor attacks. As mentioned in Section 4.1 (Experimental Setup), since BOBA aims to alter the classification results of the entire batch with only a very small proportion of samples poisoned, we cannot use a typical ASR metric. Please note that we have conducted experiments to investigate the impact of each stage in **Section 4.3 (Ablation Studies).** **Column "Stage 1" in Table 3** shows the results of a variant that directly uses the triggers derived in Stage 1 to attack the models without implementing Stage 2. The low ACR in the experimental results demonstrates the necessity of Stage 2.

---

### Official Review · Reviewer_GUp3 · 2025-10-28

**Soundness:** 3
**Presentation:** 3
**Contribution:** 2
**Rating:** 6
**Confidence:** 3

**Summary:**

This paper offers an original and thought-provoking contribution by exposing a batch-level vulnerability in BN layers and designing a corresponding attack mechanism. The experimental validation is thorough, but the clarity of presentation can be improved.

**Strengths:**

The paper identifies a previously underexplored vulnerability in BN layers under large batch settings, revealing that inter-sample dependencies can be exploited for a new type of batch-oriented backdoor attack.

The evaluation is extensive, covering multiple datasets and architectures. The introduction of new metrics like attack contamination rate (ACR) demonstrates methodological rigor.

The exploration of adaptive and differential privacy-based defenses shows a thoughtful attempt to analyze attack resistance and propose mitigation strategies.

**Weaknesses:**

While the batch-oriented perspective is interesting, the overall structure still closely parallels traditional backdoor frameworks. The novelty lies more in the attack surface than in fundamentally new techniques.

Most results are empirical. Analytical insights into why one poisoned sample can dominate batch statistics would enhance the scientific depth.

While the attack scenario differs from classical backdoors, including comparisons with state-of-the-art stealthy attacksunder modified conditions would better position the work in context.

**Questions:**

While the batch-oriented perspective is interesting, the overall structure still closely parallels traditional backdoor frameworks. The novelty lies more in the attack surface than in fundamentally new techniques.

Most results are empirical. Analytical insights into why one poisoned sample can dominate batch statistics would enhance the scientific depth.

While the attack scenario differs from classical backdoors, including comparisons with state-of-the-art stealthy attacksunder modified conditions would better position the work in context.

---

> ### Author Response · Authors · 2025-11-26
>
> **AW1:** Sorry for the confusion. We propose a novel attack paradigm with a fundamentally different purpose. BOBA aims to control the classification of an entire batch of samples, which is completely different from traditional backdoor attacks that target a single sample. To achieve this, we developed an entirely new attack method that includes a trigger derivation mechanism and a contrastive contamination-based retraining strategy, exploiting the vulnerability of BN layers.  Our work is the first to reveal a critical security risk inherent in BN layers in large batch processing scenarios, establishing a new threat beyond the traditional backdoor framework.
>
> **AW2:** Thanks for the suggestion. We added a **new Section 3.7 (Theoretical Analysis)** to the revised paper, which formally establishes the theoretical foundation of BOBA through a series of theorems, explaining why a few poisoned samples can dominate the statistics of a large batch.
>
> Specifically, **Theorem 3.1** states that Stage 1 of BOBA creates an outlier in the feature space. **Theorem 3.2** indicates that the outlier leads to a non-zero perturbation of the batch variance, enabling a deterministic shift in feature representations. Stage 2 of BOBA leverages the deterministic shift to achieve batch-oriented attacks. In this way, a few poisoned samples can manipulate the classification results of an entire batch.
>
> **AW3:** Thanks for the suggestion. We conducted new experiments to compare the performance between BOBA and SOTA backdoor attacks within our proposed attack scenarios. Specifically, we considered two advanced backdoor attack methods, i.e., **BppAttack [a] and ISSBA [b]**.
> For a fair comparison, we set the poisoning ratios ($\eta_i$ and $\eta_a$) for all baseline attacks to 10% and 1%, respectively, the same as our BOBA default settings. The track_running_stats is set to False. We used Clean Accuracy (CA) and our proposed Attack Contamination Rate (ACR) to evaluate the effectiveness of these attacks in large-batch inference (n=1024). From the following table, we find that traditional backdoor attacks have low ACR, since their design goal is not batch contamination, demonstrating the fundamental difference between BOBA, a novel batch-oriented backdoor attack, and traditional sample-oriented backdoor attacks in terms of attack goals and effects.
>
> | Attack | CA (%) | ACR (%) |
> |:---:|:---:|:---:|
> | BppAttack | 91.05 | 12.11 |
> | ISSBA | 92.26 | 11.56 |
> | BOBA | 90.31 | 82.05 |
>
> **Reference:**\
> [a] Li et al. Invisible backdoor attack with sample-specific triggers. ICCV 2021.\
> [b] Wang et al. Bppattack: Stealthy and efficient trojan attacks against deep neural networks via image quantization and contrastive adversarial learning. CVPR 2022.

---

### Official Review · Reviewer_tz6W · 2025-10-31

**Soundness:** 2
**Presentation:** 2
**Contribution:** 2
**Rating:** 4
**Confidence:** 3

**Summary:**

The paper proposed a new Batch-Oriented Backdoor Attack(BOBA), which includes two stages: trigger derivation and contrastive contamination-based retraining. As long as a carefully designed "poisoned sample" is mixed into one batch, BOBA can lead to the contamination of the prediction results of the entire batch in large-scale inference or training using Batch Normalization (BN). The experimental results show that on various models (such as ResNet, VGG, EfficientNet) and datasets (MNIST, CIFAR-10, GTSRB, Tiny-ImageNet), as long as a very small number of samples with triggers are mixed in the batch, most samples can be misclassified.

**Strengths:**

1. This paper reveals for the first time the security risk of cross-sample contamination existing in Batch Normalization (BN) in large-batch scenarios and proposes a brand-new attack view.
2. The proposed BOBA framework is divided into two stages (trigger derivation + contrastive contamination), with clear logic

**Weaknesses:**

1. The model specificity of triggers limits the generalization of this method. In this paper, triggers are derived for specific trained models, the ablation experiment also indicated that the effect of trigger derivation on untrained models was very poor.
2. Some assumptions look too strong, the first is BOBA the method need to set track_running_stats=False, when the defender set track_running_stats=True, BOBA is basically ineffective. However, in many typical deployments (especially for inference/online services), the running stats of BN is frozen (track_running_stats=True and using running_mean/var) to ensure inference stability.  The second is the attacker have a pre-trained target model (not a randomly initialized raw model, but a normally trained model with decent performance) to reverse derive the most effective trigger patch.
3. Although the paper evaluated various defenses, the defenses methods such as "noise addition" and "statistical detection" was relatively simple, how about more advanced defenses?
4. There is a lack of theoretical explanations or quantitative modeling of pollution propagation in the BN normalized equation.

**Questions:**

1. One of my concerns is that the method is only evaluated on low-resolution datasets and small models. This may limit the method's interest to a broader audience, and it is not clear why the authors chose to focus on small-scale datasets and why they did not include pre-trained models such as ViT.
2. In the experiment, the paper set batch size from 512 to 2048, how about the smaller batch size, such as 128, 256?

---

> ### Author Response · Authors · 2025-11-26
>
> **AW1:** Sorry for the confusion. Indeed, the triggers of BOBA are model-specific. However, this is exactly the core strategy of the BOBA attack, which optimizes triggers for specific models to maximize attack effectiveness. As mentioned in **Section 3.2 (Threat Model)**, attackers can access and control the target model to optimize its triggers. In this way, triggers do not need to be generalized between different models. Furthermore, the experiments in **Table 2 of the paper** show that the BOBA attack achieves satisfactory performance across different models and datasets, demonstrating strong generalization.
>
> **AW2:** Sorry for the confusion.
> Indeed, in many typical inference deployments, BN layers are often frozen. However, as mentioned in the paper, BOBA is designed for large-batch processing scenarios, where batch statistics are updated during inference to improve performance. Therefore, the threat posed by BOBA is realistic.
> Meanwhile, BOBA indeed needs a pre-trained model to derive effective triggers. However, as mentioned in **Section 3.2 (Threat Model)**, it is a realistic assumption that aligns with attack scenarios where adversaries can control the training process of a target model. In this way, accessing a pre-trained model does not impose higher capability requirements on attackers.
>
> **AW3:** Thanks for the suggestion. We conducted new experiments to evaluate BOBA's attack performance against four advanced black-box backdoor defense methods on the CIFAR-10 dataset. Specifically, we considered two detection defenses, i.e., **SCALE-UP [a] and Frequency [b]**, and two purification defenses, i.e., **ZIP [c] and SampDetox [d]**.
> From the following tables, we find that BOBA still achieves high PA and ACR against ZIP and SampDetox, demonstrating its high robustness to purification defenses.
> For SCALE-UP detection, BOBA achieves high FRR and FAR, indicating that our attack confuses the detection. Specifically, SCALE-UP amplifies the pixel values of suspicious samples by different multiples and combines them into the models to obtain their predictions. By analyzing the predictions, SCALE-UP can identify poisoned samples. However, since BOBA enables backdoor knowledge transfer among samples within a batch, it disrupts SCALE-UP sample predictions, thereby failing the SCALE-UP defense.
> For Frequency detection, we find that Frequency identifies the poisoned samples from BOBA.  This is because our work does not focus on the visual stealthiness of triggers, which results in the poisoning samples being detected, as discussed in Section 5.2 (Limitation and Future Work) of our paper.
>
> | Detection | FRR | FAR |
> |:---:|:---:|:---:|
> | SCALE-UP | 27.56 | 45.90 |
> | Frequency | 10.11 | 21.26 |
>
> | Purification | CA (\%) | PA (\%) | ACR (\%) |
> |:---:|:---:|:---:|:---:|
> | ZIP | 81.37 | 21.56 | 65.31 |
> | SampDetox | 85.24 | 41.12 | 50.26 |
>
> **Reference:**\
> [a] Guo et al.SCALE-UP: An Efficient Black-box Input-level Backdoor Detection via Analyzing Scaled Prediction Consistency. ICLR 2023.\
> [b] Zeng et al. Rethinking the backdoor attacks' triggers: A frequency perspective. ICCV 2021.\
> [c] Shi et al. Black-box backdoor defense via zero-shot image purification. NeurIPS 2023.\
> [d] Yang et al. SampDetox: Black-box Backdoor Defense via Perturbation-based Sample Detoxification. NeurIPS 2024.
>
> **AW4:** Thanks for the suggestion. We added a **new Section 3.7 (Theoretical Analysis)** to the revised paper, which formally establishes the theoretical foundation of BOBA through a series of theorems, explaining why a few poisoned samples can dominate the statistics of a large batch.
>
> Specifically, **Theorem 3.1** states that Stage 1 of BOBA creates an outlier in the feature space. **Theorem 3.2** indicates that the outlier leads to a non-zero perturbation of the batch variance, enabling a deterministic shift in feature representations. Stage 2 of BOBA leverages the deterministic shift to achieve batch-oriented attacks. In this way, a few poisoned samples can manipulate the classification results of an entire batch.

---

> > ### Author Response · Authors · 2025-11-26
> >
> > **AQ1:** Thanks for the suggestion.
> > We conducted new experiments on the **ImageNette** dataset, a subset of ImageNet comprising 10 categories and images of size 224$\times$224. From the following table, we can find that BOBA remains effective on high-resolution images.
> >
> > | &nbsp; &nbsp; &nbsp; Model | &nbsp; &nbsp; &nbsp; &nbsp; &nbsp; &nbsp; &nbsp; $n=512$ | | &nbsp; &nbsp; &nbsp; &nbsp; &nbsp; &nbsp; &nbsp; $n=1024$ | | &nbsp; &nbsp; &nbsp; &nbsp; &nbsp; &nbsp; &nbsp; $n=2048$ | |
> > | :-------------: | :-----: | :------------: | :------: | :------------: | :------: | :------------: |
> > | | Benign | BOBA | Benign | BOBA | Benign | BOBA |
> > | | CA (%) | CA(%)/ACR(%) | CA (%) | CA(%)/ACR(%) | CA (%) | CA(%)/ACR(%) |
> > | PreAct-ResNet18 | 55.96 | 51.35 / 77.73 | 54.17 | 51.62 / 74.31 | 53.26 | 50.96 / 74.79 |
> > | VGG19\_bn | 47.31 | 45.28 / 78.22 | 46.33 | 45.31 / 77.56 | 45.78 | 44.11 / 77.42 |
> > | EfficientNet-B3 | 41.56 | 37.16 / 71.19 | 40.35 | 36.46 / 70.36 | 38.66 | 35.30 / 70.51 |
> >
> > We added the above experimental results to **Table 2** in the revised paper.
> > Additionally, we added the experimental results from ablation studies and discussions on the ImageNette dataset in **Tables 3, 4, 5, 6, 7, and 8** of the revised paper, which reach the same conclusions as the original paper.
> >
> > Since the ViT model architecture utilizes LN layers rather than BN layers, BOBA cannot be applied to it at the moment. As discussed in Section 5.2 (Limitation and Future Work), exploiting the other normalization layers (e.g., LN, IN, and GN) in models to inject backdoors is a worthy area for future study. Please note that we have conducted experiments on the VGG19_bn model architecture, one of the BN-based models with the largest number of parameters, demonstrating the generalization ability of BOBA.
> >
> > **AQ2:** Thanks for the suggestion. We conducted new experiments over a larger range of batch sizes (64-8192) on the CIFAR-10 dataset. From the following table, we can find that BOBA achieves satisfactory attack performance with different batch sizes, even with small batch sizes (e.g., n=64, 128 and 256).
> >
> > | Batch size | Benign |&nbsp;&nbsp;&nbsp; &nbsp;   BOBA |
> > |:---:|:---:|:---:|
> > |  | CA (%) | CA(%)/ACR(%) |
> > | 64 | 88.18 | 88.21 / 80.02 |
> > | 128 | 91.05 | 90.17 / 82.10 |
> > | 256 | 90.33 | 89.33 / 81.43 |
> > | 512 | 92.51 | 90.22 / 82.16 |
> > | 1024 | 91.16 | 90.31 / 82.05 |
> > | 2048 | 88.37 | 88.46 / 81.90 |
> > | 4096 | 89.45 | 88.34 / 80.77 |
> > | 8192 | 90.50 | 90.31 / 81.20 |

---

### Author Response · Authors · 2025-12-03

Dear Area Chairs,

Thank you for managing the review process. We sincerely thank the reviewers for their thoughtful and constructive feedback. The rebuttal process has significantly strengthened our paper. We have addressed all concerns by providing **additional experiments** and **new theoretical analysis**. We summarize the key concerns raised by the reviewers and our corresponding responses as follows.

* **Lack of theoretical analysis (W4 from tz6W, W2 from GUp3)**\
We added **Section 3.7 (Theoretical Analysis)** to the revised paper, formally analyzed the BOBA attack mechanism using proposed Theorems 3.1 and 3.2, and explained how a few poisoned samples perturbed the batch variance and manipulated the classification results of an entire batch.

* **Lack of advanced defense methods (W3 from tz6W, W2 from D37u)**\
We conducted new experiments to evaluate BOBA's attack performance against four advanced black-box defense methods (**SCALE-UP [a], Frequency [b], ZIP [c], and SampDetox [d]**). The results show that BOBA still maintains a relatively high attack performance under most defenses.

* **Lack of experiments on high-resolution images (Q1 from tz6W, Q4 from D37u)**\
We conducted the **ImageNette** dataset, a subset of ImageNet comprising 10 categories and images of size 224$\times$224. The results show that BOBA remains effective on high-resolution images. Additionally, we added the experimental results from ablation studies and discussions on the ImageNette dataset in **Tables 3, 4, 5, 6, 7, and 8** of the revised paper, which reach the same conclusions as the original paper.

* **Lack of experiments over different batch sizes (Q2 from tz6W, W3 from D37u)**\
We conducted new experiments over a larger range of batch sizes (**64-8192**) on the CIFAR-10 dataset. The results show that BOBA achieves satisfactory attack performance with different batch sizes, even with small batch sizes (e.g., n=64, 128 and 256) or large batch sizes (e.g., n=9192).

* **Lack of evaluation on overhead (W4 and Q1 from D37u)**\
We conducted new experiments to investigate the time overhead of BOBA on the CIFAR-10 dataset. The results show that BOBA's overall time overhead is **1.177** times that of training a benign model, which is acceptable for attackers.

* **Lack of comparison with SOTA backdoor attacks (W3 from GUp3)**\
We conducted new experiments to compare the performance between BOBA and SOTA backdoor attacks (**BppAttack [e] and ISSBA [f]**) within our proposed attack scenarios. The results show the fundamental difference between BOBA, a novel batch-oriented backdoor attack, and traditional sample-oriented backdoor attacks in terms of attack goals and effects.

We believe that the revisions and additional experiments have significantly strengthened the paper, addressing all major concerns and providing deeper theoretical and empirical support for our claims. Thank you again for your time and consideration.

Best Regards,\
The authors of 15259

**Reference:**\
[a] Guo et al.SCALE-UP: An Efficient Black-box Input-level Backdoor Detection via Analyzing Scaled Prediction Consistency. ICLR 2023.\
[b] Zeng et al. Rethinking the backdoor attacks' triggers: A frequency perspective. ICCV 2021.\
[c] Shi et al. Black-box backdoor defense via zero-shot image purification. NeurIPS 2023.\
[d] Yang et al. SampDetox: Black-box Backdoor Defense via Perturbation-based Sample Detoxification. NeurIPS 2024.\
[e] Li et al. Invisible backdoor attack with sample-specific triggers. ICCV 2021.\
[f] Wang et al. Bppattack: Stealthy and efficient trojan attacks against deep neural networks via image quantization and contrastive adversarial learning. CVPR 2022.

---

### Meta-Review · Area_Chair_Ma1i · 2026-01-06

**Summary:**

The paper introduces a novel "Batch-Oriented Backdoor Attack" (BOBA), which exploits the inherent mechanism of the Batch Normalization (BN) layer. Unlike traditional backdoor attacks that target individual samples, BOBA leverages the cross-sample dependency in BN to contaminate the batch statistics (mean and variance), causing clean samples within the same batch to be misclassified when even a single poisoned sample is present. The framework operates in two stages: trigger derivation and contrastive contamination-based retraining. Experiments across multiple architectures (ResNet, VGG, EfficientNet) and datasets (MNIST, CIFAR-10, etc.) demonstrate that the attack can achieve high "Attack Contamination Rates" (ACR) even with very low poisoning budgets in large-batch settings.

**Reviewer Concerns:**

The primary concerns revolve around the practical assumptions and the scope of the evaluation. Reviewers (tz6W, D37u) pointed out that the attack's effectiveness relies heavily on the assumption that track_running_stats=False, whereas many real-world inference services freeze these statistics (track_running_stats=True) for stability, which would render the attack ineffective. Additionally, the novelty was questioned in light of existing works that exploit other intrinsic DL mechanisms (like Dropout or Down-sampling). Evaluation gaps were also highlighted: the study is confined to low-to-mid resolution datasets (missing 224x224 scale) and lacks comparisons against state-of-the-art black-box backdoor defenses (e.g., SCALE-UP, SampDetox). Finally, there were requests for more rigorous analytical modeling of how a single sample dominates batch statistics and a clearer breakdown of the computational overhead.

**Reviewer Scores:**

The paper received scores of 4, 6, and 4, reflecting a borderline consensus with a lean toward rejection. While Reviewer GUp3 (6) appreciated the originality of the batch-level vulnerability, Reviewers tz6W and D37u (4, 4) remained skeptical about the real-world threat model given the specific BN configurations required. In the rebuttal, the authors should focus on demonstrating the attack's impact under frozen BN statistics or justify why their setting is common in specific deployment scenarios. Furthermore, providing experimental results on high-resolution datasets and against SOTA defenses would be crucial for moving the borderline scores. At this stage, while the idea is thought-provoking, the dependency on strong assumptions and the limited evaluation scope lead the Area Chair to recommend rejection in its current form.

---

### Decision · Program_Chairs · 2026-01-26

Reject